# A Benchmark of Foundation Model Encoders for Histopathological Image Segmentation

**Itsaso Vitoria**[1]                                                    ITSASO.VITORIA@TECNALIA.COM
[1] *TECNALIA, Basque Research and Technology Alliance (BRTA), Derio, Spain*

**Cristina L. Saratxaga**[1]                                            CRISTINA.LOPEZ@TECNALIA.COM

**Cristina Penas Lago**[2]                                              CRISTINA.PENAS@EHU.EUS
[2] *Department of Cell Biology and Histology, University of the Basque Country, Leioa, Spain*

**Rosa Izu**[3,4]                                                       ROSAMARIA.IZU@EHU.EUS
[3] *Department of Dermatology, Basurto University Hospital, Bilbao, Spain*
[4] *Biocruces Bizkaia Health Research Institute, Barakaldo, Spain*

**Ana Sanchez-Diez**[3,4]                                              ANA.SANCHEZD@EHU.EUS

**Goikoana Cancho-Galan**[4,5]                                         GOIKOANA.CANCHOGALAN@OSAKIDETZA.EUS
[5] *Department of Pathology, Basurto University Hospital, Bilbao, Spain*

**Maria Dolores Boyano**[2,4]                                          DOLORES.BOYANO@EHU.EUS

**Ignacio Arganda-Carreras**[6,7,8,9]                                  IGNACIO.ARGANDA@EHU.EUS
[6] *Department of Computer Science and Artificial Intelligence, University of the Basque Country, Donostia-San Sebastián, Spain*
[7] *Donostia International Physics Centre (DIPC), Donostia-San Sebastián, Spain*
[8] *Ikerbasque, Basque Foundation for Science, Bilbao, Spain*
[9] *Biofisika Institute, Leioa, Spain*

**Adrian Galdran**[1,8]                                                ADRIAN.GALDRAN@TECNALIA.COM

## Abstract

Whole-slide imaging has transformed histopathology into a data-intensive field, requiring robust and generalisable computational tools. Foundation models offer a promising approach for a range of downstream tasks with minimal labelled data. While recent work has shown their effectiveness for slide-level classification and retrieval, their potential for dense prediction tasks such as image segmentation remains underexplored. In this study, we present a comprehensive benchmark of 15 pathology-specific foundation models for histopathological image segmentation, evaluated across two distinct modalities: H&E-stained histology and Annexin A5-stained immunohistochemistry. To ensure a fair and architecture-neutral comparison, we freeze each foundation models encoder and pair it with a shared lightweight decoder, disentangling representation quality from model size. Results show that foundation model encoders can sometimes lead to strong segmentation performance without fine-tuning, but effectiveness varies significantly by model and modality. Our findings reveal that compact encoders can often outperform larger, more recent models, underscoring that model size and classification accuracy are poor predictors of segmentation capabilities.

**Keywords:** Foundation models, histopathology, IHC, image segmentation.

## 1 Introduction

The increased availability of high-throughput whole-slide imaging has transformed histopathology into a data-intensive discipline. A single academic laboratory scanning surgical specimens can now produce up to 1,500 whole-slide images (WSIs) and $\sim$1.6 TB of data per day, rapidly out-scaling what pathologists can manually review or conventional pipelines can process Kelleher et al. (2023). Convolutional Neural Networks (CNNs) trained from scratch or fine-tuned on natural-image databases struggle to generalise across staining protocols or scanner vendors Tellez et al. (2019), and even across institutional biases Du et al. (2025), often requiring retraining for every new endpoint Campanella et al. (2025).

Foundation models, large neural networks pretrained on vast WSIs via self-supervised learning, address these limitations by learning domain- agnostic tissue representations adaptable to many downstream tasks with minimal labelled data Campanella et al. (2025). Recent pathology-specific models such as UNI Chen et al. (2024) and Virchow Vorontsov et al. (2024) achieved state-of-the-art accuracy on slide-level classification, retrieval and prognostic benchmarks, outperforming task-specific networks while remaining robust to cross-hospital domain shifts Xu et al. (2024). By decoupling feature learning from task-specific supervision, these models enable scalable, easily deployable computational-pathology pipelines, potentially accelerating biomarker discovery and clinical translation Wang et al. (2024).

Benchmarking efforts to date have mostly assessed foundation models on image recognition tasks. Campanella et al. (2025) compiled 22 slide-level clinical diagnostic tasks and found pathology foundation models uniformly surpassed ImageNet-trained networks on cancer detection and biomarker prediction. Breen et al. (2025) assessed 14 encoders for ovarian-tumour subtyping, again finding almost every foundation model outperforming conventional CNNs, while Lee et al. (2025) compared four domain-specific foundation models across 14 datasets under *consistency* and *flexibility* scenarios, finding that lightweight adapter tuning was sufficient to adapt them to new classification tasks. With few exceptions like Kang et al. (2023), most models treat WSIs as a set of individually annotated tiles, and evaluate global predictions resulting from this "bag" of samples at a specimen level. However, the potential of foundation model embeddings for dense prediction tasks like image segmentation remains largely unexplored.

This work closes the gap by providing the first unified head-to-head comparison of a wide array of recent pathology foundation models for segmentation. This is achieve by freezing the parameters of each foundation model encoder and then pairing them with the same lightweight decoder, which is learned from training data. In this way, representation quality, and not backbone capacity, drives our assessment, as demonstrated in Fig. 1. Comprehensive evaluation on a recently released public dataset and a proprietary database eliminate test-set leakage and reveal which foundation model embeddings are indeed useful for dense tissue delineation, providing guidance for future model selection and development.

## 2 Methodology

### 2.1 Notation and Problem Statement

**Image Tiles and Semantic Segmentation** In computational histopathology image processing tasks, whole-slide images (WSI) are typically too large to be processed end-to-

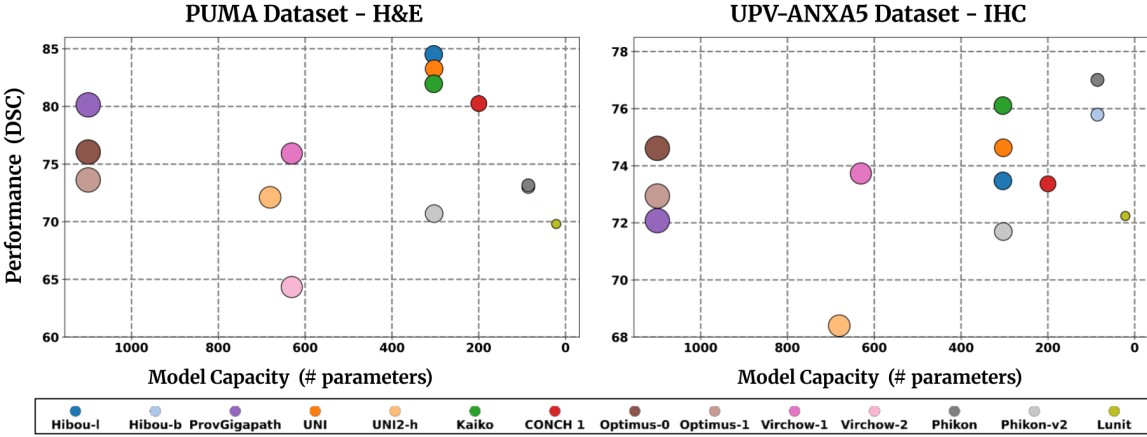

Figure 1: Using two different datasets (**PUMA**- H&E and **UPV-ANXA5**-IHC), we evaluate performance (Dice Similarity Coefficient, higher is better) vs. size (nr. of parameters, lower is better) of up to fifteen recent foundation model encoders, when repurposed for Histopathology image segmentation. Marker size reflects model capacity.

end, hence we often operate on fixed-size RGB tiles $\boldsymbol{x} \in \mathbb{R}^{H \times W \times 3}$. In semantic segmentation problems, we assume a training set of $N$ labeled tiles $\mathcal{D} = \{(\boldsymbol{x}_i, \boldsymbol{y}_i)\}_{i=1}^{N}$, where each dense mask $\boldsymbol{y}_i \in \{0, \ldots, C-1\}^{H \times W}$ assigns one of $C$ tissue classes to each pixel in a tile $\boldsymbol{x}_i$. From this data, we can then learn an encoder–decoder segmentation network $f_{\theta,\omega}$:

$$f_{\theta,\omega}(\boldsymbol{x}_i) = \hat{\boldsymbol{y}}_i = \psi_\theta\big(\phi_\omega(\boldsymbol{x}_i)\big), \quad \boldsymbol{x}_i \in \mathbb{R}^{H \times W \times 3} \;\longmapsto\; \hat{\boldsymbol{y}}_i \in \{0, \ldots, C-1\}^{H \times W}, \qquad (1)$$

with learnable encoder parameters $\omega$ and decoder weights $\theta$. At inference time, tile predictions $\hat{y}_i$ are stitched with a sliding-window strategy to recover a WSI-level mask.

**Foundation Model Embeddings** Recent work in digital pathology has produced an array of open-source foundation models, large Vision Transformers (ViT), Dosovitskiy et al. (2021), pretrained on tens of millions of histology tiles, which we will denote here as $\Phi_{\omega^\star}$. These models are often employed as powerful, general-purpose feature extractors, with their parameters $\omega^\star$ fixed ("frozen") or lightly fine-tuned.

Internally, a patch-embedding layer partitions input tiles into $P \times P$ patches ($P = 14$ or $P = 16$ in practice) and projects each patch onto a $d$-dimensional token. Each model is provided with a fixed input resolution of $512 \times 512$ pixels. The patch size $P$ is inferred dynamically from the model's architecture.

After adding positional encodings and passing through $L$ transformer blocks, the encoder outputs a sequence $\mathbf{Z}$ of $T$ tokens with low spatial resolution:

$$\mathbf{Z} = \Phi_{\omega^\star}(\boldsymbol{x}) \in \mathbb{R}^{T \times d}, \qquad T = h \times w \in \mathbb{N}, \; h = \frac{H}{P}, \; w = \frac{W}{P} \qquad (2)$$

optionally preceded by a classification (CLS) token that we discard here. Note that the embedding dimensionality $d$ is architecture dependent (*e.g.* $d=768$ for ViT-B and $d=1{,}536$

for ViT-L architectures), as is the patch size $P$. In standard multiple-instance learning pipelines, this sequence $\mathbf{Z}$ is reduced via CLS/mean/attention pooling to a single embedding $\bar{z} \in \mathbb{R}^d$ for slide-level tasks such as WSI classification, or unsupervised subject clustering.

**From Tokens to Spatial Feature Maps** Instead of pooling the sequence in eq. (2), here we restore its two-dimensional structure by a reshape and channel permuting mapping:

$$\mathbf{F} = \text{reshape}(\mathbf{Z}) \in \mathbb{R}^{d \times h \times w}, \tag{3}$$

thereby obtaining a view of the data analogous to a low-resolution *feature map* that preserves the spatial layout of the tile and is ready for decoder upsampling. In our benchmark analysis, we keep the backbone frozen and we train only a lightweight convolution-style decoder following $F$, allowing us to measure how much pixel-level information is already encoded in the embeddings produced by different foundation models.

## 2.2 Foundation Model Encoders

Our benchmarking methodology is based on the comparative evaluation of various backbone architectures in the domain of digital pathology. To ensure a robust and representative analysis, we selected a diverse set of fifteen encoder models, listed below. Further details on the architecture of the encoders can be found in the appendix A.

- **UNI** Chen et al. (2024) is a general-purpose self-supervised vision encoder pretrained using DINOv2 on Mass-100K dataset, which contains more than 100M image tiles of different resolutions, extracted from around 100,000 H&E-stained WSIs across 20 organ tissue types. **UNI2-h** is pretrained on a larger scale, using 200M image tiles extracted from more than 350,000 H&E and IHC WSIs collected from Mass General Brigham.

- **CONCH** Lu et al. (2024) is a vision-language model pretrained on 1.1M histopathology image-caption pairs available in Pubmed Central Open Access. **CONCHv1_5** is built on a ViT-L architecture initialized from the UNI pretrained checkpoint, and fine-tuned following a procedure similar to the original CONCH framework.

- **Phikon** Filiot et al. (2023) is an early ViT-B model pretrained with iBOT on over 40M image tiles from 6,000 H&E-stained WSIs from The Cancer Genome Atlas (TGCA). **Phikon-v2** Filiot et al. (2024), is a ViT-L model pretrained with DINOv2 on PANCAN-XL, an expanded dataset with 450M tiles from 55,000 H&E WSIs across 30 cancer types.

- **Virchow** Vorontsov et al. (2024) contains 632M trainable parameters and was pretrained on a 1.5M H&E-stained WSIs dataset sourced from the Memorial Sloan Kettering Cancer Center. **Virchow2** Zimmermann et al. (2024) was trained on a larger dataset of 3.1M WSIs, sampled at four different magnifications obtained from the same institution.

- **Prov-Gigapath** Xu et al. (2024) is a self-supervised model pretrained on the Prov-Path dataset, which includes 1.38B image tiles from 171,189 H&E and IHC-stained WSIs. The dataset includes 31 different tissue types, including both tumor and non-tumor tissues.

- **H-Optimus-0** Saillard et al. (2024) is a 1.1B parameter ViT trained in a self-supervised manner on >500,000 H&E-stained WSIs, including human tissues from multiple body

regions, covering 31 healthy and tumoural tissue types. **H-Optimus-1** Bioptimus (2025) is a similar model, but trained on over 1M WSIs from more than 800,000 patients.

- **Kaiko** ai et al. (2024) is a series of histopathology foundation models trained on data from the TCGA. The Kaiko family covers multiple ViT configurations, but here we analyse only the largest one (ViT-L), which performed best in our experiments.

- **Lunit**, Kang et al. (2023), is a self-supervised ViT based image classification model trained on 33M H&E-stained image tiles from multiple public datasets.

- **Hibou**, Nechaev et al. (2024), is a family of histopathology models. We consider **Hibou-b**, a ViT-B trained on 512M tiles, and **Hibou-L**, a larger ViT-L trained on 1.2B tiles. Both models are trained on a proprietary dataset including over 1M WSIs of H&E and non-H&E-stained tissues from human and veterinary sources, as well as cytology slides.

### 2.3 Encoder–Agnostic Single–Scale Decoder Design

Once an input tile $\boldsymbol{x}$ passes through a foundation model encoder $\Phi_{\omega^\star}$, we obtain a token grid $\mathbf{F} \in \mathbb{R}^{d \times h \times w}$ whose spatial resolution $(h, w)$ is a factor $P$ coarser than the original size $(H, W)$. To produce a per-pixel prediction $\hat{y} \in \{0, \ldots, C-1\}^{H \times W}$ we need a decoder mapping $\mathbf{F}$ back to the full resolution of $\boldsymbol{x}$. A natural choice is a U-Net–style, Ronneberger et al. (2015), symmetric upsampling, but this entangles decoder capacity with encoder size. Larger ViT backbones (*e.g.* ViT-L/H) would yield proportionally larger decoders than smaller variants (ViT-S/B), introducing a confounder. To isolate the *intrinsic representational quality* of encoders, we design a lightweight decoder with constant parameter count across all backbones, regardless of encoder depth or embedding dimension $d$.

Let $\mathbf{F} \in \mathbb{R}^{d \times h \times w}$ be the feature tensor produced by any encoder, where $h = \frac{H}{P}$ and $w = \frac{W}{P}$. The decoder $\psi_\theta$ is deliberately minimal and *identical* for all backbones:

1. **Width projection**: A $1 \times 1$ convolution projects the backbone–specific width $d$ to a fixed head dimension $D$, $\mathbf{F}_0 = \text{Conv}_{1 \times 1}(\mathbf{F}) \in \mathbb{R}^{D \times h \times w}$. We set $D = 256$.

2. **Progressive $\times 2$ up-sampling** Let $s = \left\lceil \log_2\left(\frac{H}{h}\right) \right\rceil$ be the number of shape doublings required to reach (or slightly exceed) the input size[1]. For $k = 1, \ldots, s$ we apply:

$$\mathbf{F}_k = \sigma\Big(\textbf{Conv}_{3 \times 3}\big(\textbf{Up}(\mathbf{F}_{k-1})\big)\Big), \tag{4}$$

where **Up** denotes $\times 2$ bilinear interpolation and $\sigma$ is a ReLU mapping. The channel width stays constant at $D$, so the complete path (projection together with $s$ upsampling blocks) always contains $\approx 2.6$ M parameters regardless of the encoder.

3. **Class logits and resize** A final $1 \times 1$ convolution yields class logits $\mathbf{L} \in \mathbb{R}^{C \times h' \times w'}$. If final upsampling exceed the target size $(h', w') \neq (H, W)$, we perform a last transformation:

$$\hat{\mathbf{y}} = \mathbf{P}\big(\mathbf{L}, (H, W)\big), \tag{5}$$

being $\mathbf{P}$ an up-scaling/down-scaling interpolation transform.

---

1. Because each decoder block doubles the spatial resolution, the natural unit for counting how many blocks we need is "how many powers of two separate the encoder grid from the input resolution".

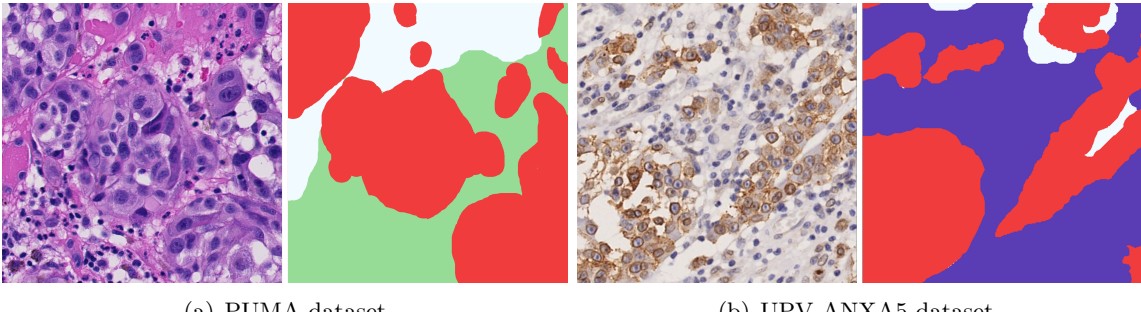

(a) PUMA dataset        (b) UPV-ANXA5 dataset.

Figure 2: Image tiles with their tissue annotation masks. (a) H&E-stained tile from the **PUMA** dataset; (b) Annexin A5-stained tile from the **UPV-ANXA5** dataset. **Red: Tumour**, **Green: Stroma**, **Blue: Tumour-Infiltrating Lymphocytes**, **White: Other**.

The encoder–decoder interface is reduced via a 1×1 projection to $D = 256$. Hence, all encoders, from ViT-S ($d = 384$) to ViT-H ($d > 1000$), supply equal-dimensional features to the upsampling head. The learnable part of the model stays fixed at just over 2.5M parameters, so any performance difference can be attributed solely to the quality of the foundation model encoder representations $\Phi_{\omega^\star}$.

### 2.4 Training Protocol

In order to enable fair comparisons, we define a common training process for all segmentation networks in our benchmark. All models are trained for 50 epochs, monitoring performance on a separate validation set each 10 epochs. Decoder weights optimize a standard linear combination of Cross-Entropy and Dice losses, using a Nesterov-accelerated Adam algorithm, with an initial learning rate set to $l = 1e\text{-}4$, cyclically annealed towards $l = 0$ each 10 epochs. We observed convergence on the training set in all our experiments.

Data is sampled in batches of 8 images and undergoes common augmentation transformations, *e.g.* random affine deformations, brightness/color jittering. We carefully normalized the intensities of input images so as to match the specifications of each foundation model encoder. Finally, the metric for both early stopping and test set evaluation purposes was the Dice Similarity Coefficient (DSC), which measures overlap between predictions and annotations. DSC was computed separately for each category and then averaged. We train each model using 10 random seeds and report average performance and standard deviations.

## 3 Experimental Analysis

### 3.1 Dataset Description

In this benchmark, two datasets from different modalities were used to evaluate the performance of the encoder models. The first is the publicly available **PUMA dataset**, Schuiveling et al. (2025), consisting of 155 primary and 155 metastatic melanoma histopathology H&E-stained $1024 \times 1024$ image tiles, scanned at $40\times$ magnification with a resolution of

Table 1: Results on the **PUMA** dataset for the Tumor, Stroma and remaining categories. Three best performances are **boldfaced**, best performance is also **underlined**.

|  | DSC-Tumor | DSC-Stroma | DSC-Other | Average DSC |
|---|---|---|---|---|
| **Hibou-l** | **92.91 ± 0.28** | 86.50 ± 0.71 | **74.16 ± 3.64** | **84.53 ± 1.30** |
| **UNI** | **93.01 ± 0.63** | **87.88 ± 0.82** | **68.96 ± 3.07** | **83.28 ± 1.30** |
| **Kaiko** | 92.12 ± 0.63 | 86.23 ± 0.69 | **67.58 ± 4.54** | **81.98 ± 1.62** |
| **CONCH 1** | **92.77 ± 0.71** | **86.98 ± 1.46** | 61.02 ± 2.76 | 80.26 ± 1.14 |
| **ProvGigapath** | 91.80 ± 0.57 | 86.21 ± 1.17 | 62.43 ± 3.56 | 80.14 ± 1.35 |
| **Optimus-0** | 91.13 ± 1.21 | 85.41 ± 1.78 | 51.64 ± 4.65 | 76.06 ± 1.95 |
| **Virchow-1** | 90.94 ± 0.83 | 84.79 ± 1.24 | 52.05 ± 6.71 | 75.92 ± 2.46 |
| **Optimus-1** | 92.61 ± 0.37 | **87.64 ± 0.54** | 40.63 ± 3.31 | 73.63 ± 1.28 |
| **Phikon** | 90.44 ± 0.78 | 85.24 ± 0.99 | 43.79 ± 7.26 | 73.16 ± 2.53 |
| **Hibou-b** | 91.49 ± 0.41 | 86.60 ± 0.70 | 41.01 ± 3.98 | 73.03 ± 1.38 |
| **UNI2-h** | 91.99 ± 0.63 | 84.78 ± 1.24 | 48.34 ± 5.52 | 72.12 ± 2.01 |
| **Phikon-v2** | 86.81 ± 1.40 | 81.67 ± 1.42 | 43.67 ± 4.06 | 70.72 ± 2.02 |
| **Lunit** | 86.62 ± 1.18 | 79.87 ± 1.58 | 42.97 ± 4.32 | 69.82 ± 1.85 |
| **Virchow-2** | 86.80 ± 2.00 | 78.70 ± 2.49 | 27.54 ± 8.34 | 64.35 ± 3.46 |
| **CONCHv1_5** | 77.16 ± 0.03 | 0.00 ± 0.00 | 5.00 ± 2.52 | 27.39 ± 0.85 |

$0.23\mu m$ per pixel (Fig. 2(a)). PUMA covers six tissue categories: tumour, stroma, epidermis, necrosis, blood vessel, and background. Since the first two classes represented more than 90% of the annotations, the remaining classes were grouped to avoid class imbalance issues impacting our analysis. The second is the **UPV dataset**, a private IHC dataset stained with Annexin A5 (ANXA5), a marker that highlights cells undergoing apoptosis by producing a brown staining signal. The intensity of this signal, along with tumour-infiltrating lymphocyte (TIL) density and morphology, can be potential biomarkers for predicting tumour recurrence and treatment response. It comprises 158 WSIs, from which 2,000 manually annotated $512 \times 512$ image tiles were extracted. Scanned at $40\times$ ($0.22\mu m$/pixel), the images were downsampled to $0.44\mu m$/pixel to match cell size in the PUMA dataset. In this case, annotations were made for Tumour, TILs and Other (Fig. 2(b)).

One of our goals is to study the generalization ability of foundation models. Therefore, we deliberately use a small number of images per modality: 15 for training, 5 for validation and 10 for test, ensuring well-balanced and representative annotation splits.

### 3.2 Quantitative Analysis

Table 1 reports Dice scores on the PUMA benchmark, averaged over three tissue categories, yielding that: **(i) Feasibility.** Even with *strictly frozen* encoders, learning only $\approx 2.6$ M decoder parameters from a modest training set yields competitive segmentation quality. **(ii) Overall ranking.** The best average DSC is obtained by **Hibou-L** (84.5%), followed by **UNI** (83.3%) and **Kaiko** (82.0%). **(iii) Class-wise trends.** Hibou-L excels on the challenging *Other* class, whereas UNI leads Dice for *Tumour* and *Stroma*, suggesting complementary spatial cues from different pre-training objectives. **(iv) Expanded train-**

Table 2: Results on the **UPV-ANXA5** dataset for the Tumor, TIL and remaining categories. Three best performances are **boldfaced**, best performance is **underlined**.

| | DSC-Tumor | DSC-TIL | DSC-Other | Average DSC |
|---|---|---|---|---|
| **Phikon** | 78.06 ± 2.07 | 68.40 ± 1.67 | **84.58 ± 0.88** | **77.01 ± 0.82** |
| **Kaiko** | 72.68 ± 1.14 | **72.36 ± 1.13** | **83.30 ± 0.9**5 | **76.11 ± 0.69** |
| **Hibou-b** | 75.92 ± 1.84 | **69.14 ± 1.51** | **82.32 ± 1.0** | **75.79 ± 0.95** |
| **UNI** | 79.56 ± 2.05 | 64.48 ± 2.68 | 79.86 ± 1.94 | 74.63 ± 1.37 |
| **Optimus-0** | 77.90 ± 1.99 | 64.79 ± 1.13 | 81.13 ± 1.32 | 74.61 ± 1.31 |
| **Virchow-1** | 74.67 ± 1.49 | 66.02 ± 1.37 | 80.49 ± 1.25 | 73.73 ± 0.97 |
| **Hibou-l** | 71.99 ± 1.02 | **68.54 ± 1.26** | 79.87 ± 1.22 | 73.47 ± 0.86 |
| **CONCH 1** | 73.49 ± 1.75 | 66.66 ± 1.47 | 79.95 ± 2.11 | 73.37 ± 1.31 |
| **Optimus-1** | **82.48 ± 2.23** | 59.11 ± 2.19 | 77.24 ± 2.02 | 72.94 ± 1.27 |
| **Lunit** | 70.94 ± 1.10 | 66.64 ± 2.09 | 79.15 ± 1.93 | 72.24 ± 1.50 |
| **ProvGigapath** | **80.62 ± 2.56** | 58.77 ± 2.93 | 76.84 ± 1.36 | 72.08 ± 1.05 |
| **Phikon-v2** | 72.75 ± 0.83 | 62.49 ± 2.24 | 79.84 ± 0.73 | 71.70 ± 0.70 |
| **UNI2-h** | **82.22 ± 1.41** | 52.74 ± 0.96 | 70.23 ± 1.27 | 68.40 ± 0.94 |
| **Virchow-2** | 69.47 ± 4.30 | 56.63 ± 0.97 | 76.23 ± 1.62 | 67.44 ± 1.36 |
| **CONCHv1_5** | 57.69 ± 3.13 | 28.32 ± 15.78 | 75.74 ± 1.70 | 53.92 ± 6.43 |

**ing does not imply performance gains.** Second-generation checkpoints with larger architectures or extended training datasets (e.g. UNI2-H vs. UNI, Virchow-2 vs. Virchow-1) underperform their predecessors, possibly because extensive classification-oriented fine-tuning weakens the positional correlations in embeddings, crucial for pixel-level tasks.

Numerical results on the **UPV-ANXA5** benchmark are reported in Table 2. Unlike PUMA, these IHC tiles differ from the H&E appearance most encoders were pretrained on. The task is therefore a good *cross-stain generalisation* test. We observe: **(i) Lower overall performance.** This dataset contains the challenging *tumour-infiltrating lymphocyte* (TIL) class, whose ambiguous borders lower DSC scores. **(ii) Phikon leads overall**, with a 77.0% mean DSC, achieving state-of-the-art performance on the *Other* class (84.6%) and competitive scores on other classes, suggesting strong generalisation. **(iii) Class-specific behaviour.** Optimus-1 excels on *Tumour*-class (82.5%) but struggles on TILs, while Kaiko attains the highest TIL Dice (72.4%). Per-class differences between these models and Phikon are noticeable. **(iv) Kaiko** is again a strong performer, ranking second Dice, with no class performance collapse. **(v) Model scale is not a guarantee.** Prov-GigaPath (>1B parameters) achieves only 42.1% average Dice, suggesting that IHC training samples did not include ANXA5-stained melanoma or that its contrastive pre-training transfers poorly to dense prediction. **(vi) Second-generation checkpoints still underperform.** As in PUMA, second-generation models (e.g. Phikon vs. v2), exhibit performance degradation.

## 4 Conclusions and Take-Home Message

In this study, we have presented a comprehensive benchmarking of fifteen foundation models for histopathological image segmentation, evaluating their performance across two dis-

tinct imaging modalities: H&E-stained histopathology (**PUMA dataset**) and Annexin A5-stained immunohistochemistry (**UPV-ANXA5** dataset). Our findings demonstrate that foundation models can help achieving strong histopathological image segmentation performance by using their frozen encoders coupled with lightweight, trainable decoders. This design allows us to isolate the intrinsic representational capacity of each foundation model encoder and assess their ability to generalize across modalities and tasks. We observed that performance varies significantly by model and modality, with pre-training data and task playing a critical role. Compact, modality-aware encoders (*e.g.*, Hibou-L, Phikon, Kaiko) often outperform larger, more computationally expensive classification-focused foundation models in dense prediction tasks (see Table 3). Furthermore, we found that second-generation models often regressed performance in our segmentation datasets, suggesting that prolonged class-level optimisation can erode the spatial correlations required for pixel-wise prediction. In general, the performance of previously reported slide-level (classification) foundation models *cannot* reflect their dense-prediction performance. In view of this, we advise practitioners to carefully benchmark foundation model embeddings for each target task, and not blindly follow a "bigger is better" model selection rule.

## Acknowledgments and Disclosure of Funding

This research has been supported by the Elkartek Programme (ONKOimaging, KK-2024/00003) and additional projects from the Basque Government (IT1524 and 20211111019). The authors are grateful to the Basque Biobank for providing the biopsy samples. A.G. is supported by grant RYC2022-037144-I, funded by MCIN/AEI/10.13039/501100011033 and co-financed by FSE+.

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

**Appendix A.**

| Name | Dataset | Dataset size | Image Modality | Embedding dim | Params | Baseline |
|---|---|---|---|---|---|---|
| UNI | Mass-100K | > 100M patches | H&E | 1024 | 303 M | ViT-L/16 |
| UNI2-h | Mass General Brigham | > 200M patches | H&E, IHC | 1536 | 681 M | ViT-H/14 |
| CONCH | PMC-OA | 1.17M image-caption pairs | H&E, IHC, special stains | 512 | 200 M | ViT-B/16 & L12-E768-H12 |
| CONCHv1.5 | PMC-OA | 1.17M patches | special stains | 768 | 307 M | ViT-L |
| Virchow | MSKCC | 1.5M WSIs | H&E | 2560 | 631 M | ViT-H/14 |
| Virchow2 | MSKCC | 3.1M WSIs | H&E | 2560 | 631 M | ViT-H/14 |
| Phikon | TCGA | > 40M patches | H&E | 768 | 86 M | ViT-B |
| Phikon-v2 | PANCAN-XL | > 450M patches | H&E | 1024 | 303 M | ViT-L |
| Prov-Gigapath | Prov-Path | > 1.4B patches | H&E | 1536 | 1.1 B | ViT |
| H-Optimus-0 | Proprietary | > 0.5M WSIs | H&E | 1536 | 1.1 B | ViTG/14 |
| H-Optimus-1 | Proprietary | > 1M WSIs | H&E | 1536 | 1.1 B | ViT |
| Kaiko | TCGA | 29k WSIs | H&E | 1024 | 304 M | ViT-L/14 |
| Lunit | Multiple | 33M patches | H&E | 384 | 22 M | ViT-S/8 |
| Hibou-b | Proprietary | 512M patches | H&E, no-H&E, cytology | — | 86 M | ViT-B/14 |
| Hibou-l | Proprietary | 1.2B patches | H&E, no-H&E, cytology | 1024 | 304 M | ViT-L/14 |

Table 3: Overview of the features of the evaluated encoders. Abbreviations: PMC-OA, Pubmed Central Open Access; MSKCC, Memorial Sloan Kettering Cancer Center; TCGA, The Cancer Genome Atlas

