# OpenReview forum: "A Benchmark of Foundation Model Encoders for Histopathological Image Segmentation"
_MICCAI.org/2025/Workshop/COMPAYL — COMPAYL 2025_

### Official Review · Reviewer_5xbY · 2025-07-12
**Foundation models are great for MIL, but what about segmentation?**

**Rating:** 5
**Confidence:** 5

**Review:**

### Summary

This paper benchmarks all publicly available foundation models (FMs) for computational pathology on two segmentation tasks—one involving H&E-stained images and the other using IHC-stained images.


### Strengths

- While the problem may not appear immediately compelling, it addresses a significant and practical challenge. Foundation models often struggle with dense prediction tasks, and this study demonstrates that this underperformance is consistent across various FMs. Notably, some models perform significantly worse than others. I found it particularly interesting that newer versions of the same model can underperform compared to earlier versions—likely because these models are not optimized for image prediction tasks (e.g., H-Optimus v0 vs. v1).
- The paper is generally well-organized, with a clear explanation of the models and appropriate dataset selection.

### Weaknesses

- The study would benefit from a broader set of benchmarks to reinforce its conclusions. For instance, including tasks beyond image segmentation—such as nuclei classification—could provide a more comprehensive evaluation of FM performance in computational pathology.

---

### Official Review · Reviewer_fFtV · 2025-07-13
**A Benchmark of Foundation Model Encoders for Histopathological Image Segmentation**

**Rating:** 2
**Confidence:** 4

**Review:**

## Summary
The paper conducts a comprehensive benchmark of 15 pathology-specific foundation models for histopathological image segmentation, focusing on H&E-stained histology and Annexin A5-stained immunohistochemistry. The authors freeze the encoders of these models and pair them with a shared lightweight decoder to ensure a fair comparison. The results demonstrate that while some foundation model encoders achieve strong segmentation performance without fine-tuning, their effectiveness varies significantly across models and modalities. Notably, smaller, more compact encoders often outperform larger, more recent models.

- **Clarity**: The paper is well-organized, but the discussion of key findings (e.g., the underperformance of certain models like CONCHv1-5) is insufficient, leaving critical questions unaddressed.
- **Quality**: The choice of encoders is well-suited.The reliance on a single metric (Dice score) and lack of external validation limits the robustness of the evaluation.
- **Originality**: The benchmark itself is valuable, but methodological novelty is lacking.
- **Significance**: The findings contribute to the understanding of foundation models in pathology, particularly by highlighting the limitations of larger models in segmentation tasks.

## Strengths
- **Comprehensive Model Selection**: The study includes all relevant computational pathology encoders, providing a thorough comparison.
- **Evaluation on IHC Data**: The inclusion of immunohistochemistry (IHC) data is a meaningful contribution, as this modality is often underrepresented in benchmark studies.
- **Fair Comparison Framework**: The use of a shared lightweight decoder ensures a consistent evaluation across models.

## Weaknesses
- **Lack of External Validation**: The study does not include external validation datasets, which would strengthen the generalizability of the findings.
- **Limited Metric Selection**: The reliance on the Dice score alone is insufficient for a robust evaluation of segmentation performance.
- **Unexplained Model Performance**: The poor performance of CONCHv1-5 on certain metrics (e.g., DSC-Stroma and DSC-Other for PUMA) is not adequately discussed, raising concerns about potential issues with the model or evaluation setup.
- **Information Bottleneck Bias**: The projection of high-dimensional encoder embeddings into a lower-dimensional decoder space may introduce an unfair advantage for smaller models, potentially explaining why they outperform larger ones. An ablation for the decoder embedding dimension would have been valuable.

---

### Official Review · Reviewer_9ygC · 2025-07-14

**Rating:** 5
**Confidence:** 4

**Review:**

**Short summary**

This paper presents the first comprehensive benchmark of pathology foundation models for semantic segmentation across two common histopathology modalities: H&E and IHC. To ensure fair comparison, the authors design a lightweight and fixed decoder architecture that maps tile-level features from foundation models to dense segmentation maps. They show that models with stronger classification performance or larger parameter count do not necessarily yield better segmentation results, as measured by the Dice similarity coefficient.

**Strengths**

- the paper addresses an important topic: benchmarking foundation models for segmentation in pathology, which remains largely under-explored compared to classification benchmarks
- the study is exhaustive and includes a wide range of foundation models
- the use of a lightweight & identical decoder across all models is key to isolate and measure the representation quality of the foundation model benchmarked
- the use of a proprietary dataset helps avoid contamination from pretraining on public datasets, allowing for a fair comparison of foundation models segmentation capacities

**Weaknesses**

- while limiting training and validation data size is reasonable, expanding the test set would strengthen the robustness of the conclusions
- some implementation details are missing, especially regarding image resolution and the patch size used for each model (see comments below) : this information is necessary to assess whether models are being evaluated under optimal conditions

**Detailed comments**

1- what resolution (in μm/px or mpp) are images processed at? are these resolutions sufficient to resolve the structures you aim to segment, especially in IHC?
2- did you account for different input resolutions across foundation models (e.g. Virchow expects 224x224 tiles at 0.5 mpp while CONCH expects 448x448 tiles at 0.5 mpp)?
3- during inference, do you use overlapping sliding windows? if so, how are overlapping predictions merged together?
4- in §3.1, you mention standard data augmentations during training. I assume embeddings recomputed at each iteration for each augmented image: what is the time breakdown between embedding and decoder training?
5- in §3.2, last paragraph: (a) please add a reference to Table 3 for clarity; (b) the sentence “More in general, previously reported slide-level (classification) foundation model performance cannot predict dense-prediction performance” is confusing ; rephrasing it would help